

**Spatially explicit global gross domestic product (GDP) data**
**set consistent with the Shared Socioeconomic Pathways**
Tingting Wang[1], Fubao Sun[1,2,3,4*]
1. Key Laboratory of Water Cycle and Related Land Surface Processes, Institute of
Geographic Sciences and Natural Resources Research, Chinese Academy of Sciences,
Beijing, China
2. State Key Laboratory of Desert and Oasis Ecology, Xinjiang Institute of Ecology and
Geography, Chinese Academy of Sciences, Urumqi 830011, China
3. Akesu National Station of Observation and Research for Oasis Agro-ecosystem,
Akesu, China
4. College of Resources and Environment, University of Chinese Academy of Sciences,
Beijing, China
**Corresponding Author:** Fubao Sun (Sunfb@igsnrr.ac.cn), from Key Laboratory of
Water Cycle and Related Land Surface Processes, Institute of Geographic Sciences and
Natural Resources Research, Chinese Academy of Sciences





## Abstract

The increasing demand of ScenarioMIP is calling for GDP projections of high resolution for the future Shared Socioeconomic Pathways (SSPs) in both socioeconomic development and in climate change of adaption and mitigation research. While to date the global GDP projections for five SSPs are mainly provided at national scales, and the gridded data set are very limited. Meanwhile, the historical GDP can be disaggregated using nighttime light (NTL) images but the results are not open accessed, making it cumbersome in climate change impact and socioeconomic risk assessments across research disciplines. To this end, we produce a set of spatially explicit global Gross Domestic Product (GDP) that presents substantial long-term changes of economic activities for both historical period (2005 as representative) and for future projections under all five SSPs with a spatial resolution of 30 arc-seconds. Chinese population in SSP database were first replaced by the projections under the two-children policy implemented since 2016 and then used to spatialize global GDP using NTL images and gridded population together as fixed base map, which outperformed at subnational scales. The GDP data are consistent with projections from the SSPs and are freely available at http://doi.org/10.5281/zenodo.4350027 (Wang and Sun, 2020). We also provide another set of spatially explicit GDP using the global LandScan population as fixed base map, which is recommended at county or even smaller scales where NTL images are limited. Our results highlight the necessity and availability of using gridded GDP projections with high resolution for scenario-based climate change research and socioeconomic development that are consistent with all five SSPs.



## 1 Introduction

The development of socioeconomic projection scenarios plays a key role in the assessment of climate change impact and socioeconomic risks for the coming decades (O'Neill et al., 2014; Wilbanks and Ebi, 2014). The Shared Socioeconomic Pathways (SSPs), which qualitative and quantitative describe broad patterns of possible global socioeconomic development with assumptions about climate change and policy responses under different challenges to mitigation and adaptation (O'Neill et al., 2014), are one of the core contents in the Intergovernmental Panel on Climate Change (IPCC) scientific assessment reports (IPCC, 2014) and in the current literature (O'Neill et al., 2016; Wilbanks and Ebi, 2014). The climate projection scenarios in Scenario Model Intercomparison Project (ScenarioMIP) are formed based on different SSPs corresponding to specific representative concentration pathways (RCPs) within Phase 6 of the Coupled Model Intercomparison Project (CMIP6) (O'Neill et al., 2016). Scenarios of future socioeconomic impact on the global environment are built upon projections of economic output and strongly require socioeconomic data support of higher spatial resolution for the coming decades (B. Merz et al., 2010; O'Neill et al., 2016; Wilbanks and Ebi, 2014).

The Gross Domestic Product (GDP) is a standard indicator to assess and compare economic development within and across countries (Kummu et al., 2018; Nordhaus, 2011; Tobias, 2018), and is usually collected at national scale (Tobias, 2018). However, the collection of official GDP data at a finer resolution (e.g., at state, city or county levels) is problematic, especially in many developing countries (Kummu et al., 2018; Nordhaus, 2011). It is crucial to spatialize GDP data into a fine-scale so that it can be easily integrated with data from other disciplines (Chen et al., 2020; Kummu et al., 2018; O'Neill et al., 2016). A growing number of openly available historical GDP data sets are provided with the development of satellite-derived nighttime light (NTL) images and gridded population to support current research at various spatial scales in a more convenient way (Bennett and Smith, 2017; Doll et al., 2006; Ghosh et al., 2010; Nordhaus, 2011; Zhao et al., 2017). The Defense Meteorological Satellite Program's



Operational Linescan System (DMSP-OLS) NTL imagery has been successfully used
in GDP redistribution for 1992-2013. However, the disaggregated GDP depends highly
on the DN values where a certain number of saturated pixels exist in DMSP-OLS NTL
images, resulting in underestimations in urban centers and overestimations in rural
regions (Zhu et al., 2017) but can be revised when incorporating with other ancillary
data like gridded population (Zhao et al., 2017). The global Soumi National Polar-
Orbiting Partnership Visible Infrared Imaging Radiometer Suite (NPP-VIIRS) NTL
imagery made up this saturation problem and advanced its calibration, providing a more
accurate approach in GDP downscaling since 2012 (Bennett and Smith, 2017). More
research on reduction in exposure and vulnerability and increase in resilience to climate
extremes can benefit from a spatially explicit GDP data set with increasing precision of
NTL image products and population count at grid level (Chen et al., 2017; Chen et al.,
2020; Wang et al., 2019; Wilbanks and Ebi, 2014).
However, the widely used GDP projections in the SSP database were provided only
at national and super-national scales from several global institutes, which have depicted
a wide range of uncertainty within different organizations (Riahi et al., 2017) and
limited the usage of integration with data from other disciplines. Moreover, the spatially
explicit global gridded GDP projections for all five SSPs are very limited, and
Murakami and Yamagata (2019) have downscaled the global population and GDP for
SSP1-3 only. Worse still, most socioeconomic development indicators for like the total
factor productivity, capital stock, and labor input etc., are either short of data sources or
provided mainly at national scale without proper conditions to make spatially explicit
GDP predictions for future scenarios. The increasing vulnerability, exposure and
resilience of socioeconomic activities to climate extremes are driving a need to move
beyond administrative unit-based analyses to enable flexible integration with datasets
of spatially explicit population and economic activities of long-term SSPs (Chen et al.,
2017; Jones et al., 2015; Su et al., 2018; Winsemius et al., 2016).
Government policy change has a strong effect on GDP and should be taken into
account for credible and quantitative information on demographic changes and
socioeconomic development (Huang et al., 2019). The one-child for each couple policy

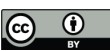



implemented in China since the late 1970s has been replaced by the two-children policy
since 2016, and would no doubt have a substantial effect on the demographic
composition, the total population and GDP projections in China in the long run.
However, this policy was implemented after the release of population and GDP
projections in the SSP database. Jiang et al., (2017; 2018) have updated Chinese
population and GDP projections at provincial level that qualitatively consistent with
five SSP narratives, showing that the implementation of two-children policy can
mitigate the labor shortages and aging problems in China to a certain extent, and are
expected to a 38.1 – 43.9% increase in GDP in the late 21st century (Huang et al., 2019).
It would be beneficial to update SSP database of long-term demographic and economic
projections in China with consideration of this two-children policy for future GDP
downscaling for spatial analyses.
To date, there is no global gridded GDP for all five SSPs provided, and historical
dataset are mainly based on national GDP from the World Bank and then redistributed
using NTL images with other auxiliary information but are not open accessed. There is
a growing demand for spatially explicit GDP that can represent different patterns of
development and are consistent with all five SSPs to match the ScenarioMIP research.
The objective of this study is to present a set of spatially explicit global GDP that
presents substantial long-term changes of GDP for both historical period (2005 as
representative) and for future projections under all five SSPs by incorporating various
data sources and methods. In the following were the inputs, assumptions,
methodologies, and results that we use to spatialize GDP data into a fine-scale,
providing an alternative choice for scenario-based climate change research and
socioeconomic development pathways.

**2 Data**
**2.1 Historical Population**

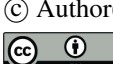



The distribution of population count (density) is a core indicator in measuring, mapping and assessing the exposure, vulnerability, and resilience of socioeconomic activities to climate extremes (Leyk et al., 2019). Several well-known global population data sets, namely the Gridded Population of the World (GPW), the Global Rural Urban Mapping Project (GRUMP), the WorldPop, and the LandScan Global Population database are summarized in this section.

### 2.1.1 the GPW Dataset

Using the areal interpolation techniques, the Gridded Population of the World dataset, Version 4 (GPWv4), Revision 11, was constructed from national or subnational administrative units in conjunction with the most detailed spatial resolution available from the Population and Housing Censuses occurring in 2005 and 2014. After extrapolated to produce population estimates for the years 2000 to 2020 at a 5-year interval with a resolution of 30 arc-seconds (approximately 1 km at the equator), these estimates were further adjusted to national totals to consist of the United Nation's World Population Prospects (UN-WPP) adjusted population estimates and densities for those years. The GPW dataset includes estimates for 2000, 2005, 2010, 2015 and 2020 respectively, and is freely accessible at http://sedac.ciesin.columbia.edu/data/collection/gpw-v4. The GPWv4 has provided globally consistent and spatially explicit disaggregated population data that is compatible with data set from other disciplines.

### 2.1.2 the GRUMP dataset

The Global Rural-Urban Mapping Project, Version 1 (GRUMPv1) dataset, which is based on GPWv3, has well identified urban area with observations of NOAA's NTL data collected over several decades. It differs from GPW by incorporating urban-rural reallocation of spatially distributed population in each census unit, and contains eight global data sets: population count, population density, urban settlement points, urban-extents, land/geographic unit area, national boundaries, national identifier, and



coastlines. GRUMPv1 provides global population estimates for 1990, 1995, and 2000
at a resolution of 30 arc seconds (approximately 1 km at the equator) as well as at
national, continental, and global scales at
https://sedac.ciesin.columbia.edu/data/set/grump-v1-population-density. The GRUMP
was the first global database that connects NTL images with population estimates, and
helps better understand differences between urban and rural areas in terms of
vulnerability, exposure, and resilience to environmental and climate change.

### 2.1.3 the WorldPop dataset

Growing from the AsiaPop, AfriPop, and AmeriPop population mapping projects,
the WorldPop (www.worldpop.org) was initiated in Oct 2013 and provided full open
access archive of spatial demographic information around the world (Stevens et al.,
2015; Tatem, 2017). Based on the random forest model and contemporary census data
from hundreds of national statistics offices and other organizations, survey, remote
sensing outputs and geospatial data etc., the WorldPop produces consistent gridded
population density at 3 and 30 arc-seconds (about 100 m at the equator for individual
countries, and about 1 km for the global mosaics, respectively). Then it was adjusted to
match the official United Nations population estimates for 2000 to 2020 annually.
Comparing with previous gridded population results, the WorldPop shows clear
advantage in its method advancement, contemporary and easily-updatable consistent
population distribution, characteristics and changes over time, enabling flexible
integration with datasets on other types of geospatial data.

### 2.1.4 the LandScan dataset

The LandScan Global Population database from the Urban Oak Ridge National
Laboratory, USA is a widely used population data set that developed using best
available census and geographic data, remote sensing imagery analysis techniques
within a multivariate dasymetric modeling framework to disaggregate census counts
within an administrative boundary (Bhaduri et al., 2007). Commercial data was utilized



in LandScan for higher spatial accuracy in population allocation at 30 arc-seconds
resolution for 1998, and 2000-2018 annually. The DN values represent population totals
per grid cell. The global LandScan population data set is now available to the
educational community free of charge at https://landscan.ornl.gov/, and has also been
widely used in GDP disaggregation with fine reliability.

### 2.1.5 census

National and subnational (at state level) population totals around the global can be
easily obtained from the World Bank. Meanwhile, census at county level for the U.S.
and China were adopted from the U.S. Census Bureau and the Statistical Yearbook from
National Bureau of Statistics, China (NBS) respectively, offering the flexibility to
perform analysis at state (or provincial) and county levels.

## 2.2 GDP

The official GDP is usually collected at national scale, but it is often problematic
to obtain data at a finer resolution (e.g., at state, city and county levels), especially in
many developing countries (Nordhaus, 2011). Using NTL images and some other
auxiliary data can help improve the quality of spatial allocation of GDP and offer a
reliable substitution to conduct cross-disciplinary research a large literature.

### 2.2.1 national and subnational GDP

The World Development Indicators assembled by the World Bank (WB-WDI)
provide a vast resource of relevant, high-quality, and internationally comparable
socioeconomic statistics for 217 economies and more than 40 country groups which
can be tracing back to more than 50 years. For most economies, GDP PPP (GDP
converted using Purchasing Power Parity rates) values are extrapolated from the 2011
International Comparison Program (ICP) benchmark estimates or imputed using a



statistical model based on the 2011 ICP. National GDP (in PPP) and GDP per capita
(Pcap) figures in 2005: in current U.S. dollars and in current international dollars, were
chosen and to be consistent with currency unit of GDP for SSP scenarios. For the
meantime, national population totals are obtained from WB-WDI database as well.
Subnational GDP in 2005, 2010 and 2015 for the U.S. and China were obtained
from departments of the U.S. Bureau of Economic Analysis and the Chinese National
Bureau of Statistics, offering the flexibility to socioeconomic performance at state (or
provincial), city and county levels. The Chinese GDP were obtained from the China
Statistical Yearbooks, the China City Statistical Yearbooks, and the China County
Statistical Yearbook with values recorded in RMB currency unit and then converted to
USD using conversion factors provided from the World Bank.
**2.2.2 the NTL-based GDP**
The NTL images have shown well correlation with global and regional economic
activities and been widely used to spatialize GDP data into a fine-scale (Ghosh et al.,
2010; Nordhaus, 2011). The widely used version 4 DMSP-OLS stable NTL images for
1992 - 2013 can be obtained from the National Oceanic and Atmospheric
Administration's National Geophysical Data Center (NGDC) at
https://www.ngdc.noaa.gov/eog/dmsp/downloadV4composites.html, with a spatial
resolution of 30 arc-seconds and latitudinal and longitudinal extent from 75˚N to 65˚S
and 180˚W to180˚E. There are two separate annual stable NTL images derived from
two sensors to avoid degradation problem, and each stable NTL image is a composition
of all the available cloud-free data with background noises and ephemeral lights
removed within this year. The DN values for DMSP-OLS stable NTL data range from
0 to 63 with saturation problem (DN values of 63) scattered mainly in city centers and
other brightly lit zones (Bennett and Smith, 2017). To improve the DMSP-OLS data
quality, the new generation of NTL products, namely the Suomi-NPP-VIIRS Day/Night
Band (DNB) images, were lunched since 2012 with a higher resolution of 15 arc-
seconds and a wider radiometric detection range. The Suomi-NPP-VIIRS DNB data



can be obtained from https://eogdata.mines.edu/download_dnb_composites.html.
The NTL images have been widely used in spatial allocation of GDP at resolutions
from 500 m × 500 m to 1° × 1° (Bennett and Smith, 2017; Nordhaus, 2011; Zhao et al.,
2018). Based on the theory that national and sub-national GDP totals are directly
distributed or regression related to each pixel in proportion to the DN values, global
and regional NTL-based GDP can be spatialized into a fine-scale and integrated with
data across disciplines (Chen et al., 2020; Ghosh et al., 2010; Zhao et al., 2017; Zhu et
al., 2017).

## 2.3 SSP projection data

The long-term demographic and GDP projections have been promoted by different
organizations to facilitate research on future impacts, adaptation, and vulnerability. The
five SSPs (O'Neill et al., 2014), which are differentiated by different combinations of
climate change mitigation and adaptation challenges, provide a wide range of
information on possible global socioeconomic developments decennially up to 2100.
SSP1 ("Sustainability") characterizes a world shifting gradually but pervasively in a
sustainable path with low mitigation and adaptation challenges, emphasizing more on
human well-being than economic growth. SSP2 ("Middle of the Road") follows a path
of continuing historical trends associated with moderate income growth, facing medium
challenges to mitigation and adaptation. SSP3 ("Regional Rivalry") is characterized
by slow economic growth with restriction of high mitigation and adaptation challenges.
SSP4 ("Inequality") represents a highly unequal world with high adaptation challenges,
and economy growth rate inclines more to rich countries. Finally, SSP5 ("Fossil-fueled
development") characterizes a world of rapid economic growth with high mitigation
challenges.

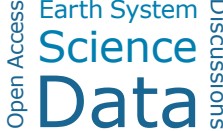

### 2.3.1 SSP Database


Based on harmonized assumptions for the interpretation of the SSP storylines in
terms of the main drivers of economic growth, three sets of global GDP projections
were provided in June 2013 in the SSP database. Recommended by the SSP database,
the Organization for Economic Cooperation and Development (OECD) (Dellink et al.,
2015) has developed a set of GDP projections based on different perspectives on future
socioeconomic development, emphasizing on the key drivers of economic growth in
the long run: population, total factor productivity, physical capital, employment and
human capital, and energy and fossil fuel resources for 184 OECD countries up to the
end of 21st century in 2005 USD in PPP for five SSP scenarios. The other two sets of
GDP projections were developed by the International Institute for Applied Systems
Analysis (IIASA) (Cuaresma, 2015) for 144 countries, and the Potsdam Institute for
Climate Impact Research (PIK) (Leimbach et al., 2015) for 32 world regions. These
three sets of GDP projections are openly available from the SSP database hosted by the
IIASA Energy Program at https://tntcat.iiasa.ac.at/SspDb.
The three sets of GDP projections were developed using the demographic
projections to maintain consistency in assumptions with education and ageing but
differed with respect to the employed drivers, methodology and outcomes, spanning a
wide range broadly representative of the current literature, and inevitably subjecting to
large uncertainties especially for the later decades (Riahi et al., 2017). A wide range of
possible factors, like policy actions, external shocks, governance barriers, and
feedbacks of greenhouse gas emission and climate extremes, are failed to predict and
disregarded in the SSP framework. Whatsoever, these GDP projections do illustrate a
substantial variance in global socioeconomic development and provide a basis for
quantitative analyzing the climate change impacts on economic for each SSP (O'Neill
et al., 2014; Riahi et al., 2017).
Together with the GDP projections, the long-term demographic projections (KC
and Lutz, 2017) in the SSP database for each SSP scenario were developed by the IIASA
and the National Center for Atmospheric Research (NCAR). Using a multidimensional



demographic model, national populations were projected based on alternative
assumptions on future fertility, mortality, migration and educational transitions in each
country for five SSPs (O'Neill et al., 2014; Riahi et al., 2017). The population
projection can well capture the link between human capital and income growth in the
econometric model as highlighted in the literature (KC and Lutz, 2017), and can also
be accessed at https://tntcat.iiasa.ac.at/SspDb.

### 2.3.2 population projection of 1/8 degree from SEDAC

No doubt that the national projections in the SSP database failed to meet the
increasing demand of spatially explicit demographic and GDP projections. Hence,
Jones and O'Neill (Jones and O'Neill, 2016) have further extended the national totals
and produced a scenario-based gridded population data set by downscaling the urban
and rural population projections for each of the 232 countries to a spatial resolution of
1/8° (approximately 7.5 arc-minutes at the equator). The gridded population from the
GRUMP dataset in 2000 at a resolution of 2.5′ was used as the base-year population for
future projection downscaling. Using the parameterized gravity model-based approach,
the demographic driving factors, namely the current fertility, income, urbanization, and
international migration are explicitly included in the population projections
corresponding to each SSP and thus can reflect its spatial pattern as prescribed. The
gridded population projections data set are quantitatively consistent with total, urban,
rural populations at a national level at ten-year intervals for 2010-2100, and with
urbanization projections as well as with the assumptions of SSP narratives.

### 2.3.3 GDP projections from NIES, Japan

Based on Jones and O'Neill, Murakami and Yamagata (2019) from the Center for
Global Environmental Research, National Institute for Environmental Studies (NIES),
Japan, have developed a new set of data by spatializing the national population and
GDP into 0.5-degree grids for SSP1, SSP2 and SSP3. As described in Murakami and



Yamagata (2019), this gridded population projection data set trumps data from Jones
and O'Neill by utilizing not only the urban and nonurban populations, but also taking
the intensity of interactions among cities and auxiliary variables including road network,
land cover, and location of airports into account. Among which, national urban
populations are downscaled into cities based on a city growth model, and then used to
project urban expansion/shrinkage with help of those auxiliary data. The GDP
projections were then developed based on its populations at a spatial resolution of 0.5-
degree for SSP1-3 only as well.
However, only two years data from settlement Points, v1 from GRUMP, SEDAC
(http://sedac.ciesin.columbia.edu/data/set/grump-v1-settlement-points) were used in
the city growth model parameterization and then in the future urban
expansion/shrinkage projections (Murakami and Yamagata, 2019), which would raise
some doubt on its credibility in these gridded population and GDP projections data sets.

### 2.3.4 Chinese Population Projections under two-children policy

The implementation of two-children policy in 2016 and clear differences among
the drivers (e.g., age, education enrollment in the historical period) between the NBS,
China and the U.N. (Huang et al., 2019) require updates in Chinese population and GDP
projections for all five SSPs. Since the demographic changes play a decisive role on
future labor force and therefore affecting socioeconomic development, Jiang et al.
(2017; 2018) have adopted data from China Statistical Yearbook and the Sixth National
Population Census and made projections of Chinese population and GDP for 2020-2100
based on assumptions of future fertility, mortality, and migration for each of five SSPs
under two-children policy. Using the parameterized population-development-
environment analysis model, Jiang et al. (2017) have implemented the population
projections incorporating with national and provincial age, sex, and educational
attainment that are quantitatively consistent with changes of birth rate under two-
children policy in China. The provincial population projections were provided at a
spatial resolution of 0.5-degree for five SSPs.



Chinese GDP (Jiang et al., 2018) were also projected using the total factor
productivity, capital stock, and labor force etc., as input for five SSPs but not been
utilized in this research.

## 3 Method

### 3.1 the official data interpolation

National GDP PPP (in USD) for 2005 were obtained from the WB-WDI first for
the 189 countries provided (data in China was for mainland only, and GDP in Hong
Kong Special Administrative Region, Macao Special Administrative Region and
Taiwan were divided but listed as individuals). For 36 island countries like British
Virgin Islands, Cayman Islands, Cook Islands and etc., where GDP were unavailable
from WB-WDI, data were obtained from the Central Intelligence Agency (CIA) World
Factbook (released in 2015). GDP in Taiwan (China), Nauru, and Syrian Arab Republic
were from the International Monetary Fund (IMF, released in Oct 2017). GDP for the
rest 11 countries (namely Aland Islands, French Guiana, Holy See, Curacao, Bonaire
Saint Eustatius and Saba, Norfolk Island, Pitcairn, Saint-Barthelemy, South Sudan,
Svalbard and Jan Mayen Islands, and Tokelau) were set as zero where no data to be
found. All GDP data were presented in PPP in 2005 USD using conversion factors
provided from the World Bank.
Meanwhile, census data were also obtained to calculate GDP per capita in order to
spatially allocate global GDP. National population totals in 2005 were obtained from
the WB-WDI for 216 countries and regions. For the rest countries (regions) without
available official census, data from other organizations were used instead. Population
in Taiwan (China) was obtained from IMF. For Norfolk Island, Pitcairn, Saint-
Barthelemy, Svalbard and Jan Mayen Islands, Guernsey, and Jersey, their population
were obtained from CIA World Factbook. Mayotte, the Holy See, Cook Islands,
Falkland Islands (Malvinas), French Guiana and others, 17 countries in total,



populations were obtained from the Wire & Plastic Products Group. Population in
Aland Islands was obtained in ASUB National accounts since it is not provided in those
global organizations.
The official GDP for China and the U.S. in 2005 were obtained the China Statistical
Yearbooks and the BEA at state (or provincial) and county levels. GDP for 51 states
and 3193 counties in the U.S., and for 31 provinces and 2326 counties with valid values
in China were obtained for validation and further updates. Meanwhile, global census at
state level (1865 states with valid values) were obtained from World Bank as well.

## 3.2 population-based GDP disaggregation

### 3.2.1 baseline population selection

The historical gridded population varies substantially since they differ in the
reliability and variety of input data sources, the interpolation and decomposition
methods of disaggregating national and subnational totals, the modeling approach, and
how they cooperate with each other to determine population distribution. To choose
more suitable gridded population as base map in GDP disaggregation, comparisons
were made between national and subnational census against the gridded data sets from
the GPWv4, the GRUMP, the WorldPop and the LandScan, which were spatially joined
to the corresponding GIS-based administrative boundaries.
At national and state scales, the biases were all relatively small in 2005. The $R^2$ are
all approaching to 1.0 and averaged RMSE are 1.2, 4.1 and 5.1 million people for the
GPWv4, World Pop and LandScan at national scale (Figure S1(a)). For 1565 states
(provinces) around the globe, their $R^2$ are around 0.98 and averaged RMSE are 1.7, 1.8
and 1.8 million, respectively (Figure S1(b)). Further comparison of 3193 counties in
the U.S. in four selected years: 2000, 2005, 2010 and 2015 showed that, $R^2$ are all
around 0.95 and their slope are approximately 0.99 including about 200 specific
counties with clear biases population totals. This is obvious since the openly available





census are the primary input in constructing these gridded data sets.

<Figure 1>

Comparisons at county level in China for the years of 2000 (1870 out of 2345
counties with valid values), 2005 (1874 counties), 2010 (1923 counties) and 2015 (1801
counties) (Figure 1) show that the LandScan outperformed in its accuracy with $R^2$
slightly higher and RMSE relatively smaller, approximately two thirds of RMSE from
the GPWv4 and the WorldPop (Figure 1). This shows that the Landscan can well
estimate population redistribution at county level than GPWv4 and WorldPop, and
therefore recommended as base map in spatial allocation of global GDP.
**3.2.2 Population Based GDP disaggregation**
Population can well capture the link between human capital and income growth in
the econometric model, and broad literature has emphasized the role of human capital
as a key driver of economic growth (Cuaresma, 2015; Dellink et al., 2015; KC and Lutz,
2017). Shiogama et al. (2011) have suggested the robustness of an ensemble learning-
based downscaling approach, which are defined by (baseline variable) × (control
variable) in accordance with distribution weights. This approach can be applied in
spatial allocation of global GDP based on the LandScan population (Pop$_{pixel}$, as baseline
variable) and GDP per capital (Pcap, ratio of GDP to population totals in a given
administrative boundary i, as control variable) to 1 km × 1 km grids (denote GDP$_{Pop}$).

$$GDP_{Pop} = Pop_{pixel} \times Pcap = Pop_{pixel} \times \frac{GDP_i}{Pop_i} \qquad (1)$$


### 3.3 NTL involved GDP disaggregation

#### 3.3.1 NTL-based GDP disaggregation

The satellite-derived NTL data has been proven to correlate well with GDP at all examined scales and has been widely used in spatial allocation of GDP over large areas (Ghosh et al., 2010; Nordhaus, 2011). The DMSP-OLS NTL images in 2005 (average visible, stable lights, and cloud free coverages, satellites F14 and F15 simultaneously collected global NTL images and data from F15 was chosen as newer sensor would have less degradation of data quality) have been utilized to disaggregate global GDP to a spatial resolution of 30 arc seconds. Based on the theory that the GDP totals are directly distributed to each pixel in proportion to the DN values in a given administrative boundary, the NTL-Based GDP disaggregation (denoted $GDP_{NTL}$) can be described as,

$$GDP_{NTL} = GDP_{per\_light} \times DN_{pixel} = \frac{GDP_i}{SL_i} \times DN_{pixel} \qquad (2)$$

where $GDP_i$ is the GDP totals, $SL_i$ is the sum of DN values and $GDP_{per\_light}$ is the constant in administrative unit i, $DN_{pixel}$ and $GDP_{pixel}$ are the DN value and corresponding GDP in each pixel in administrative unit i.

#### 3.3.2 NTL & population based GDP disaggregation

The saturation problem in the DMSP-OLS NTL images, however, has resulted in overestimation in urban centers and underestimation in rural and distanced areas. Zhao et al., (2017) have improved its accuracy by incorporating the gridded population data into NTL-based GDP disaggregation in each pixel since population data has an exponential relationship with DN values of NTL images. By multiplying the NTL image with the LandScan population data in 2005, Lit-Pop image was produced and then used in Equation 3 to spatialize GDP at global scale (denoted $GDP_{Lit-Pop}$):

$$GDP_{Lit-Pop} = \frac{GDP_i}{SLP_i} \times DN_{lp} \qquad (3)$$

where $DN_{lp}$ is the DN value of each pixel of Lit-Pop data, and $SLP_i$ is the sum of the



DN values of Lit-Pop image in administrative unit i.

## 3.4 Historical GDP disaggregation

To examine the performance of three GDP disaggregation approaches above,
namely the $GDP_{Pop}$, $GDP_{NTL}$ and $GDP_{Lit-Pop}$, national GDP in China and U.S. from WB-
WDI instead of official state or county values were used to spatialize global interpolated
official GDP into 1 km×1 km grid using the global LandScan population, DMSP-OLS
NTL images in 2005. Meanwhile, GDP PPP (in 2005 USD) from 52 states in USA plus
31 provinces in China, 321 cities in China, and 3068 plus 2091 counties in USA and
China in 2005 have been adopted and spatially joined to the corresponding GIS-based
administrative boundaries respectively, and used to verify the disaggregated GDP
results.

<Figure 2>

The comparisons showed that the accuracy of three disaggregated GDP decreases
accompanied by the changes of their spatial scales, and $GDP_{NTL-Pop}$ is superior to
$GDP_{Pop}$ and $GDP_{NTL}$ at national, state (provincial), and county levels with clear
advantages evaluated by their $R^2$ and RMSE. In detail, $GDP_{Lit-Pop}$ can better identify the
spatially allocated GDP at finer spatial scales with higher accuracy with $R^2$ reaching
0.78 and RMSE of 8.35 billion USD for 5221 counties in the U.S. and China. While the
$R^2$ is only 0.47 and RMSE reaches as high as 14.34 billion between official GDP and
$GDP_{NTL}$, indicating less advantageous due to the saturation problem.
Meanwhile, GDP using the global LandScan population only as base map ($GDP_{Pop}$)
can well identify GDP redistribution at finer spatial scales as well. The $R^2$ between
official GDP and $GDP_{Pop}$ at county level in the U.S. and China reaches as high as 0.73
and the averaged RMSE is 9.80 billion USD in 2005, which performs better than
$GDP_{NTL}$ and is nearly comparable to $GDP_{Lit-Pop}$, indicating that $GDP_{Pop}$ can be used as



an alternative when $GDP_{Lit-Pop}$ is limited.
Similar results can be obtained in 2015 as another validation case when using the
NPP-VIIRS NTL images, the global LandScan population and official GDP (Figure S2).
National GDP from WB-WDI were used in GDP disaggregation, and subnational GDP
from the U.S. Census Bureau and the corresponding Statistical Yearbooks in China
were used for validation purpose. $GDP_{Lit-Pop}$ outperformed with higher $R^2$ of 0.96 in 52
states in U.S. plus 31 provinces in China, and their RMSE (90.39 billion USD) is about
one-half of that of $GDP_{Pop}$, and only one-third of $GDP_{NTL}$, showing clear advantage in
spatial allocation of GDP at a medium spatial scale. Meanwhile, $GDP_{Pop}$ and $GDP_{Lit-Pop}$
both outperformed than $GDP_{NTL}$ at county level, and $GDP_{Pop}$ even performs a slightly
superior with smaller RMSE of 14.11 billion USD than $GDP_{Lit-Pop}$ (RMSE of 15.36
billion USD) (Figure S2).

All above showed that population involved base map can be used in spatial
allocation of GDP as well. The $GDP_{Lit-Pop}$ is recommended for global, state and county
scales disaggregation, and $GDP_{Pop}$ can be used as an alternative and especially at county
or even smaller scales where NTL images are limited in very rural regions.
Based on above, we updated official GDP and GDP per capita (in PPP) at county
level in the U.S. and China, and then disaggregated the global GDP in 2005 into 1 km
×1 km grid based on the NTL images and the LandScan population together as base
map to ensure spatial accuracy. This gridded GDP PPP in 2005 was used as historical
gridded GDP in the following comparison. $GDP_{Pop}$ in 2005 was also provided.

## 3.5 Global GDP downscaling for SSPs

NTL image projections for future scenarios are off limits and therefore unavailable
for spatial allocation of GDP for different SSP scenarios. Luckily, a set of global
spatially explicit population projections that are consistent with SSPs was developed
with a spatial resolution of 0.125 degree (Jones and O'Neill, 2016), which can be used



in spatial allocation of GDP projections to a spatial resolution of 0.125 degree first using
the above $GDP_{Pop}$ approach. Assuming that there will be no population mobility and
such within countries and across the grids, the GDP projections can be further
downscaled to 1 km × 1 km grids using NTL images and global LandScan population
in 2015 as fixed base map.
First, we completed the GDP and population projections for all the countries
(regions) in the SSP database. Population projections from IIASA were adopted since
its historical data in 2005 were less biased with $R^2$ approaching 1.0 and averaged bias
of 0.27% for 178 countries when compared against national population totals from WB-
WDI data set. Population (GDP) projections for 182 (177) countries were obtained from
IIASA (OECD, as recommended) in the SSP database. Meanwhile, the supranational
projections for the rest countries where the associated world regions the countries
belong to were obtained and filled to complete the future time series to ensure the
consistency for all five SSP scenarios.
Next, we recalculated the GDP per capital. Instead of using the exact GDP (PPP in
2005 USD), population, and GDP per capital predictions directly since these values
vary greatly among different originations, GDP per capita growth rate relative to that
in 2005 (provided from SSP database) were obtained for each country. The new national
GDP per capita for all five SSPs were recalculated by multiplying these growth rates
with GDP per capita settled (based on the WB-WDI) for 2005. Meanwhile, extra
arrangement was set for the following countries. To be more specific, in Pitcairn, Saint
Helena, Svalbard and Jan Mayen Islands, and Tokelau, the national GDP per capital
growth rate were set as 1.0 (constants for future scenarios due to missing predictions)
for all five SSPs. The Somalia's GDP per capita growth rate was assumed to follow that
of the African region since its GDP was missing in 2005. Furthermore, GDP per capital
growth rate in Switzerland and Sudan were used to replace the values in Liechtenstein
and South Sudan instead of the regional data due to geopolitical reason.
Then we disaggregate the global GDP projections using $GDP_{Lit\text{-}Pop}$ approach. We
first updated Chinese population in the gridded population of 1/8 degree from SEDAC

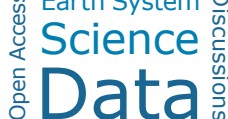

538 with population projections developed by Jiang et al., (2017) under two-children policy

539 in China, which were downscaled to a spatial resolution of 0.125 degree for all five

540 SSPs. After national and regional GDP per capita recalculated by utilizing the above

541 GDP per capita, and spatially joined to the corresponding administrative boundaries,

542 national GDP were preliminary redistributed by multiplying with scenario-based global

543 population projections (Jiang et al., 2017; Jones and O'Neill, 2016) with a spatial

544 resolution of 1/8° for 2030-2100 at 10-year intervals for all five SSPs using the $GDP_{Pop}$

545 approach. The DMSP-OLS stable NTL data in 2013 was adopted to replace the negative

546 DN values from the Suomi-NPP-VIIRS DNB images in 2015. After resampled to a

547 spatial resolution of 1 km, the global LandScan population in 2015 were introduced to

548 calculate the base map. Following the $GDP_{Lit-Pop}$ approach, the preliminary

549 redistributed GDP at 1/8° resolutions were further disaggregated to a spatial resolution

550 of 30 arc seconds (~1 km) for all five SSPs, using Lit-Pop in 2015 as fixed spatially

551 explicit pattern of GDP. Spatially explicit global GDP in 2005 and in 2030, 2050, and

552 2100 (as representative) are shown in Figures 3-5 to present substantial long-term

553 changes of GDP under five SSP scenarios.

554

555  <Figure 3>

556  <Figure 4>

557  <Figure 5>

558

559  Last, we disaggregate the global GDP using LandScan population only as base map

560 as an alternative choice. Following the same procedure, the LandScan global population

561 in 2018 (latest obtained) was used as base map, and the above preliminary global GDP,

562 which were downscaled to a spatial resolution of 1/8° for 2030-2100 at 10-year intervals

563 for all five SSPs with Chinese GDP projections updated under the two-children policy,

564 were disaggregated to 1 km×1 km grids (2030, 2050, and 2100 as representative and

565 shown in Figure S6-S8). The GDP projections based on $GDP_{Pop}$ approach can be used

566 as an alternative when NTL images are limited in very rural regions or at a finer spatial



scale.
It is worth mentioning that the LandScan population data set was used as base map
as an alternative in GDP disaggregation as 1) using population data set as base map
performs no worse than that of GDP$_{Lit-Pop}$ (Figure 2), and 2) valid values only exists
when the original NTL images and population were both not null in Lit-Pop, and that
may result in some overestimation in city area.

## 4 Result

Consistent with the national totals in the SSP database and the SSP narratives,
global and regional GDP depict different patterns among different SSP scenarios. The
highest GDP projection will reach more than 21 times in SSP5 while the lowest
projection only stays around 4.4 times in SSP3 that of 2005 by 2100 at global scale.
Visible differentiations appear around 2060 with averaged about 4.9 times that of 2005
but expand to about 4.4 - 12.8 times by 2100 for SSP1-SSP4 globally. GDP in all five
SSPs depict varying degrees of development with a slowing down in GDP growth rates
over time, especially in the second half century in most developing countries.
Meanwhile, GDP projections vary greatly across nations but are mainly consistent with
the national GDP growth rate projections from the SSP database. For example, GDP in
the U.S. expands only about 4.8 times in SSP5 and to about 2.2 times in SSP3 that of
2005 by 2100.
By replacing with two-children policy, the GDP projections in China, however, has
led to different growing pattern among SSP scenarios. It exhibits a persistent increasing
trend with highest of about 9.7 - 40.6 times that of 2005 by 2100 for all five SSP. While
GDP projection from the SSP database shows a rapid development with a peak of
around 2070-2080 for SSP1 and SSP3-5 with highest rates of about 7.1 - 18.7 times
that of 2005 and then declined to about 6.9 - 18.1 times by 2100. These differences of
Chinese GDP are result from the change of population due to the two-children policy,



which are predicted to continue growing with a peak of approximately 1.39 - 1.45
billion around 2030, and then to decline under four SSPs with the exception of SSP3
(Jiang et al., 2017), against the continue growing with a peak of 1.36 - 1.40 billion
around 2030 and then to decline under all five SSPs in the SSP database.
The regional GDP also depicts major differences inequality. Taking Northeast
America (including Virginia, West Virginia, Pennsylvania, Connecticut, Delaware,
Maryland, New Jersey, New York, and District of Columbia), five countries in Europe
(including Netherlands, Germany, Belgium, France, and Luxembourg), and Circum-
Bohai Sea Region in China (including Beijing, Tianjin, Hebei, Liaoning, and Shandong
provinces) as case study since these three regions share similar latitude, highly
developed, and are densely populated areas. Their GDP vary substantially among
different SSP scenarios as well as among different regions over time (Figure 6), with
highest growth rate reaching about 5.3, 5.2, and 39.2 times (in SSP5) but lowest of
about 2.4, 2.5, and 9.4 times (in SSP3) that of 2005 by 2100 for five countries in Europe,
Northeast America and the Circum-Bohai region in China, respectively. European
region and Northeast America show similar GDP growth rate over time, and the city
centers and places along traffic show much higher GDP (about 50 to 100 billion USD
in per grid) than rural regions (less than 5 billion) in these three regions (Figure 6).

<Figure 6>

**5 Data availability**
There are two sets of global GDP (PPP in 2005 USD to enable comparison among
years and across regions) disaggregation results for 2005 as historical period and for
2030-2100 as future projections for SSP1-5 at 10-year interval provided, one with Lit-
Pop in 2015 as base map and the other using LandScan population in 2018 as base map.
The two data sets are provided in "tif" format with a spatial resolution of 30 arc-seconds
(approximately 1 km at the equator). The global GDP are disaggregated within its

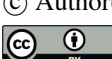



administrative boundaries, and therefore the Antarctica, oceans as well as some desert
or wilderness areas are filled with value 0. The spatial extents are 65S-75N and 180E-
180W (limited due to the Suomi-NPP-VIIRS NTL image extent), and 55.875S-83.65N
and 180E~180W in standard WGS84 coordinate system for two data sets, respectively.
The detailed information regarding to these GDP disaggregation results is available
from "Global dataset of gridded GDP scenarios", which is provided by the Global
Change Risk of Population and Economic Systems (GCR-PES): Mechanisms and
Assessments Project, Beijing Normal University, Beijing, China
(http://gcr.bnu.edu.cn/). The two sets of gridded GDP projections are available at
https://doi.org/10.5281/zenodo.4350027 (Wang and Sun, 2020).

## 6 Discussion and conclusion

In this study, we produced a set of spatially explicit global GDP, which to the best
of our knowledge, the first data set that presents substantial long-term changes of GDP
for both historical period (2005 as representative) and for future projections under all
five SSP scenarios with a spatial resolution of 1 km. The combination of gridded
population and NTL images outperformed in GDP disaggregation across the globe, and
official census and GDP in U.S. and China at county level were incorporated within
GDP disaggregation. Chinese population in SSP database were replaced by Jiang et al.,
(2017) which incorporates data from China Statistical Yearbook and the Sixth National
Population Census at provincial scale and may offer a higher precision, and then used
to spatialize GDP under two-children policy. The main objective is to provide a set of
spatially explicit global GDP projections that is readily applicable across disciplines,
and $GDP_{Lit-Pop}$ is recommended at national, state and county scales, while $GDP_{Pop}$ is
recommended at county or even smaller scales where NTL images are limited in very
rural regions.
However, this GDP dataset was bound to the national and subnational data of
various data sources, and to the approaches including using uniformed national GDP

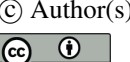



per capita growth rate within a country, using fixed gridded population and NTL images
in specific historical year as base map for future GDP disaggregation, and etc.
First, the national and super-national population and GDP in SSP database are
highly depend on the methodology used in projection, including the model, the input
drivers, and assumptions of future developments, leading to varying projections from
different global organizations. Similar to the vast majority of literatures, the effect of
financial crisis and climate change policies, scientific and technological progress, and
many political and societal factors are, however, in absence beyond those in place when
data was developed for GDP disaggregation. The climate system feedbacks are not
considered on GDP disaggregation for five SSPs as well. The uncertainties for original
SSP projections, especially where data coverage is limited, also exist in this
disaggregated GDP and should be treated with caution.
Second, using fixed spatial distribution of gridded population and NTL images at
historical level as base map is based on the assumption that population mobility within
countries and across the grids will not occur, thus the gridded GDP projections fail to
capture the future spatial differences caused by population migration. Meanwhile, the
DN value of zero in either gridded population or NTL images (e.g., regions like farther
north of 65N or very rural places) can directly cause zero proportion of GDP, resulting
in some bias in such regions (GDP downscaling using the LandScan population as only
base map is recommended as an alternative).
Last, simple approach of using uniform national GDP per capita growth rate within
a country to downscale the national GDP to match the future population totals at 0.125
degree, can cause an even distribution of GDP in space, and is highly correlated with
projected population distribution. Other inevitable shortages in this approach, like using
the existing data that are combined with various techniques to replace missing values
for future scenarios, the currency conversion factors used at national scale and etc., are
no doubt adding more uncertainly in both historical and future GDP disaggregation.
Despite various known shortcomings and uncertainties that discussed above, this
gridded GDP data set can provide a chance to allow for comparability of global and
regional socioeconomic changes between historical period and future projections under



different socioeconomic development pathways as described by the SSPs. It can also
broaden the applicability of regional economic activities and potentially feed back to
climate impact research. Our results highlight the necessity and availability of using
gridded GDP projections with high resolution, especially in hazard exposure,
vulnerability, and resilience analysis for the ScenarioMIP research.

**Author contributions.**
TW and FS designed the research, and TW performed the analysis and drafted the
manuscript; FS provided insights on data product characteristics and underlying
procedures.

**Competing interests.**
The authors declare that they have no conflict of interest.
**Acknowledgements**
This research was supported by the National Key Research and Development
Program of China (2016YFA0602402, 2019YFA0606903), the TopNotch Young
Talents Program of China (Fubao Sun), China Postdoctoral Science Foundation funded
project (2018M640173, 2020T130646) and the National Natural Science Foundation
of China (42001031).

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

**Figure Captions:**
Figure 1 Comparisons between official census and the gridded extractions from global
population data sets for the years 2000 (a), 2005 (b), 2010 (c) and 2015 (d) at county
level in China.

Figure 2 Comparison between official and disaggregated GDP at national level (a), and
at state (b) and county (c) levels in U.S. and China in 2005, values in brackets are the
RMSE.

Figure 3 The spatial allocation of global GDP using GDP$_{Lit-Pop}$ approach for 2005 (a)
and 2030 under SSP1-5 scenarios (b-f) at a spatial resolution of 1 km.




Figure 4 The spatial allocation of global GDP using GDP$_{Lit-Pop}$ approach for 2005 (a)
and 2050 under SSP1-5 scenarios (b-f) at a spatial resolution of 1 km.

Figure 5 The spatial allocation of global GDP using GDP$_{Lit-Pop}$ approach for 2005 (a)
and 2100 under SSP1-5 scenarios (b-f) at a spatial resolution of 1 km.

Figure 6 The spatial allocation of GDP in selected regions (Northeast America (a series),
five countries in Europe (b series), and Circum-Bohai Sea Region in China (c series))
for 2005 as historical period and for 2030, 2050, and 2100 using GDP$_{Lit-Pop}$ approach
under SSP1 scenario as study case (1 km resolution). Their spatial distribution and
corresponding regional GDP growth (times that of 2005) are in the bottom.




**Figure**

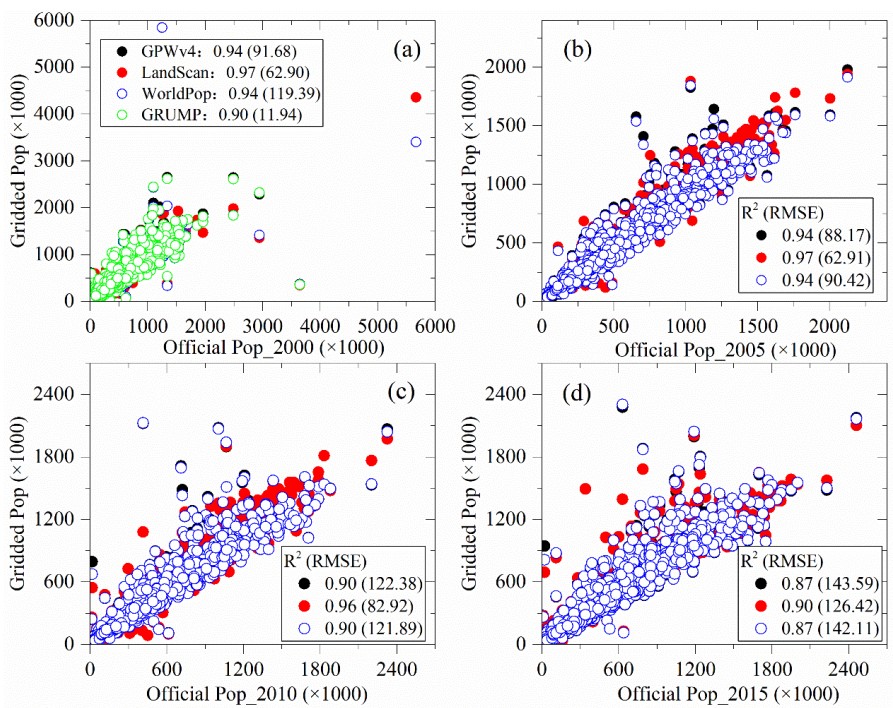

Figure 1 Comparisons between official census and the gridded extractions from global
population data sets for the years 2000 (a), 2005 (b), 2010 (c) and 2015 (d) at county
level in China.

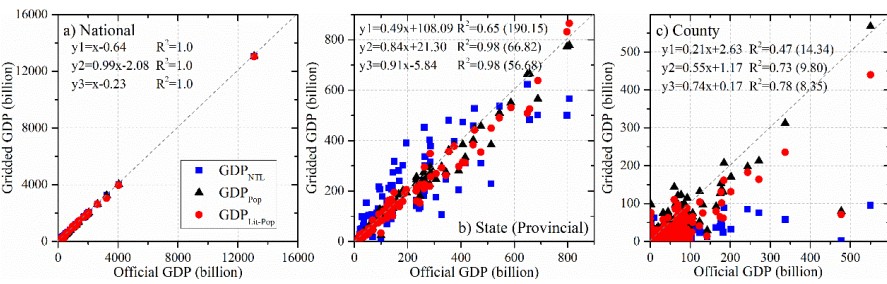

Figure 2 Comparison between official and disaggregated GDP at national level (a), and
at state (b) and county (c) levels in U.S. and China in 2005, values in brackets are the
RMSE.



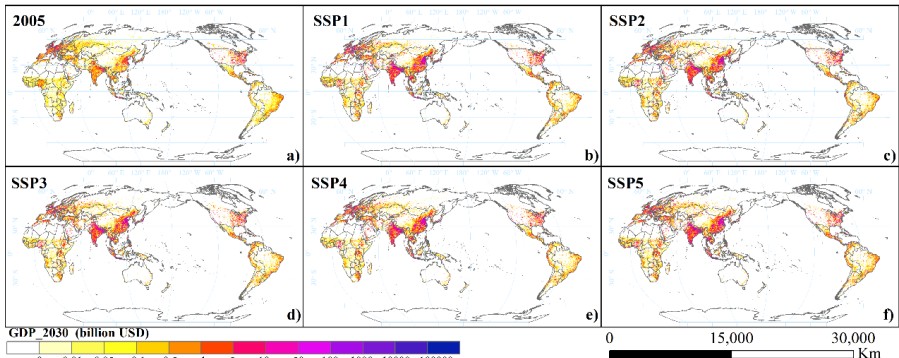

Figure 3 The spatial allocation of global GDP using GDP$_{Lit-Pop}$ approach for 2005 (a) and 2030 under SSP1-5 scenarios (b-f) at a spatial resolution of 1 km.

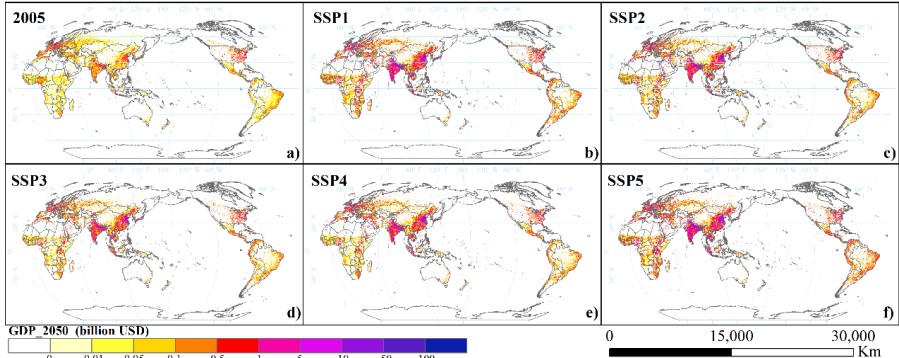

Figure 4 The spatial allocation of global GDP using GDP$_{Lit-Pop}$ approach for 2005 (a) and 2050 under SSP1-5 scenarios (b-f) at a spatial resolution of 1 km.

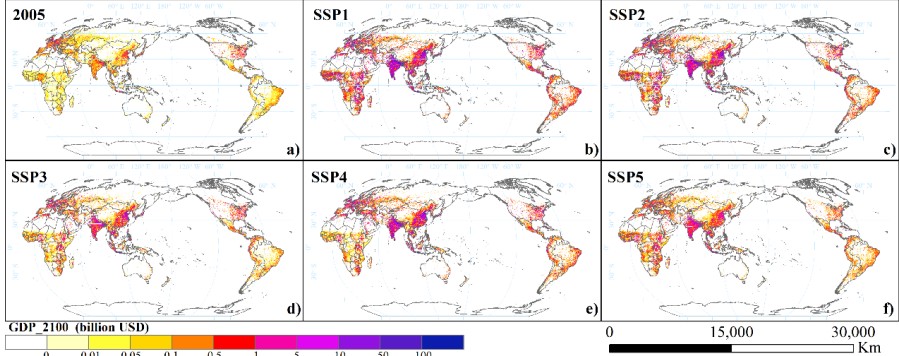

Figure 5 The spatial allocation of global GDP using GDP$_{Lit-Pop}$ approach for 2005 (a) and 2100 under SSP1-5 scenarios (b-f) at a spatial resolution of 1 km.

859

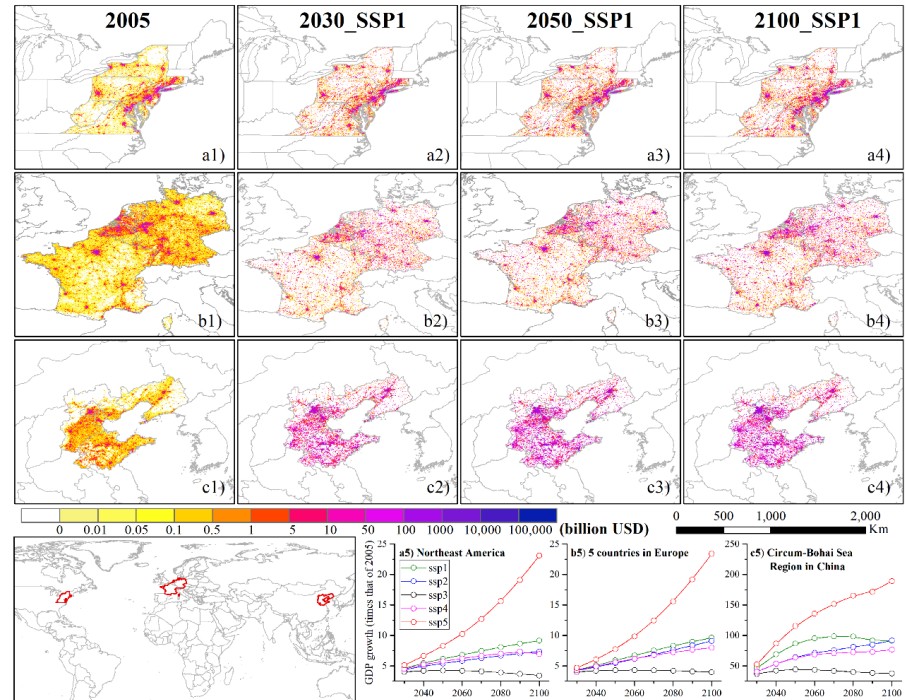

860

Figure 6 The spatial allocation of GDP in selected regions (Northeast America (a series), five countries in Europe (b series), and Circum-Bohai Sea Region in China (c series)) for 2005 as historical period and for 2030, 2050, and 2100 using GDP$_{Lit-Pop}$ approach under SSP1 scenario as study case (1 km resolution). Their spatial distribution and corresponding regional GDP growth (times that of 2005) are in the bottom.



