# Peer review of "Spatially explicit global gross domestic product (GDP) data set consistent with the Shared Socioeconomic Pathways"

_Earth System Science Data, 2021_

## Author Comment (AC2)

**To: Referee #2, ESSD**

**Subject:** Revise the manuscript (#essd-2021-10)

**The Authors:** Tingting Wang, Fubao Sun*

**The Title:** Spatially explicit global gross cell product (GCP) data set consistent with the Shared Socioeconomic Pathways

**Major Comments**

Q1) Title: The authors name their data spatially explicit gross domestic product (GDP), which is very colloquial and misleading as GDP usually refers to the output per country and not per cell. Consider using gross cell product instead.

Thank you for this suggestion. Done.

Q2) Introduction: The introduction is not very well structured and needs revision: E.g. lines 84-88 appear very unconnected to the preceding paragraphs, as are lines 98-102.

Thank you for this suggestion. The introduction has been largely improved. First, the original lines 84-88 have been removed from this manuscript, and lines 98-102 have been revised and moved to a more proper position. Then, we have added some literature review on GDP related data set, which was unaware of or missed by accident before, like Geiger et al. (2017) and Gao (2017) as suggested. Besides, the limitations of NTL-based and population-based GDP disaggregation approach have been addressed in the introduction as well. Other minor revisions have been made and won't be elaborated here.

Q3) Line 74-75: The authors should cite and refer to this recent paper, which is an global extension of Zhao et al 2017 and covers large parts of the methodological work presented in the current manuscript in much more detail: Eberenz, S., Stocker, D.,

Röösli, T., and Bresch, D. N.: Asset exposure data for global physical risk assessment, Earth Syst. Sci. Data, 12, 817–833, https://doi.org/10.5194/essd-12-817-2020, 2020.

Thank you for recommending this work from Eberenz et al. (2020). The improved LitPop approach is exactly the approach that we are looking for in GDP disaggregation, or else we wouldn't provide a population-based GCP data set in our first version. It has been cited, described (section 3.3), evaluated (section 3.4), used (sections 3.5 and 3.6) and discussed (section 6) in our revised manuscript. Really helps a lot.

Q4) Line 94-95, line 317-334: Murakami and Yamagata downscaled SSP1-3 only for a specific reason: They found their very elaborate methodology to be too uncertain to be applied to SSP4-5. In the current manuscript I am missing a thorough discussion of related uncertainties.

Thank you for pointing this out, and we felt curious before why projections for SSP1-3 only were provided. Since their data set was not used in ours, we have revised as *"Murakami and Yamagata (2019) used a series of auxiliary variables, and projected and downscaled the global population and GDP to 0.5-degree grids for SSP1-3 with explicitly consideration of spatial and socioeconomic interactions among cities, while data for SSP4 and SSP5 was not available."* The original section 2.3.3 with its detailed description has been removed from this manuscript.

Q5) Line 118-120: This statement about missing open access availability is not true. There are various freely available sources for gridded historical and/or future GDP data, e.g. Eberenz et al 2020, Geiger et al (2017), Kummu et al (2018).

Sorry about this mistake.

We have revise this as *"To date, global GCP projections for all five SSPs with a high resolution is rather limited, and historical GCP was bound to national GDP of various data sources and to the disaggregation approach with clear limitations of population or NTL images as described above."*

Q6) Section 2 (starting from line 131) is a presentation of various data sources without any introduction whatsoever. It would clearly be helpful for the reader to provide an initial paragraph motivating the description and the use of the different data sources in the context of the study.

Thank you for this suggestion, and we have added a paragraph before section 2.1.

*"The pursuit of reliable GCP data set lies in higher accuracy of inputs and proper downscaling approach. In the following, we descripted the data, including gridded population and NTL images used for base map generation, official GDP and GRP, and GDP projections that would be used in GDP disaggregation."*

Q7) Section 2.2.1 (starting from line 205) is very confusing and it remains totally unclear why the concept of the ICP or PPP rates are introduced. This, however, is a very relevant point fundamental for the whole study! SSP data is provided in units of 2005 PPP USD (to allow for global intercomparison of national numbers) but most other data is not (or not anymore) (E.g., Worldbank GDP data in 2005 PPP USD was withdrawn and I don't expect sub-national GDP data to be available in PPP-adjusted values). Please see Geiger (2018) and references therein for a thorough description of this issue. I expect the authors to describe in clear language including references to data sources how they tackled the problem of data conversion step by step. It's not a simple issue and faulty data might be the result. This comment also applies to Section 3.1.

Sorry about this mistake, we have reproduced this GCP data from the beginning. National GDP of 195 countries in 2005 PPP USD was obtained from Geiger et al. (2017) first. Meanwhile, GRP in 2005 for 394 OECD large (TL2) regions from 36 countries was obtained from OECD. Meanwhile, GRP in 31 provinces and 2160 counties with valid values in China, and valid figures in 51 states and 3071 out of 3145 counties in U.S.A were obtained from China Statistical Yearbooks and the U.S. Bureau of Economic Analysis.

These GRP figures are in current PPP USD. So we first calculating the ratio between GDP in 2005 PPP USD (Geiger et al., 2017) and GDP in current PPP USD (obtained from World Bank) in each country. Then these GRP can be rescaled to 2005 PPP USD in each national boundary i using a simple method as below with regional inflation differences being overlooked.

$$GRP_{2005} = \frac{GDP_{i\_2005}}{GDP_{i\_current}} \times GRP_{current}$$

National GDP for 36 OECD countries, USA and China were replace by 394 TL2 regions and over 5000 counties. Among which, GRP in Virginia, USA instead of county figures was used since GRP of 50 counties in Virginia was missing, and same was for Tibet Autonomous Region, China. Meanwhile, reaggregated GRP base on $GDP_{LitPop}$ during validation procedure (section 3.4) was adopted and converted to 2005 PPP USD for about 150 counties in China and USA since their official GRP was missing. Finally, these GDP and GRP figures in 2005 were used as input for GDP disaggregation.

A more detailed data and method description is in sections 2.3 and 3.5 in our revised manuscript.

Thank you.

Q8) Section 2.3.2 (starting from line 301): SEDAC for a couple of years already provides downscaled SSP population projections at 1km scale (https://sedac.ciesin.columbia.edu/data/set/popdynamics-1-km-downscaled-pop-base-year-projection-ssp-2000-2100-rev01). These data are not mentioned neither applied in the current manuscript and could significantly improve the results provided.

Thank you for this suggestion. This SSP population projections at 1 km resolution has been mentioned and cited in our revised introduction. This SEDAC_1km population has been used in LitPop approach, and the resulted GCP produced is shown in right column, which shows more rasterized at smaller scales. The GCP is more spatial explicit when using LandScan as population base map. After comparison (below, not mentioned in the manuscript) and much consideration, we are more prone to use SSP

population projections of 0.125 degree and downscaled into 1 km grids to then applied to LitPop approach.

[Figure]

Figure a1. Comparison of GCP for 2030 under SSP1 scenario, using LandScan as base map in SSP population projection at 0.125-degree downscaling (a1 and a2), and using SSP population projection at 1-km directly (b1 and b2), as inputs in LitPop approach. The above is English Channel, and the bottom is southern China.

Q9) Line 317-334: A revised version of the Murakami data was provided at 1/8° resolution (using the Jones and O'Neill 2016 population data as input) here: Geiger, Tobias; Daisuke, Murakami; Frieler, Katja; Yamagata, Yoshiki (2017): Spatiallyexplicit Gross Cell Product (GCP) time series: past observations (1850-2000) harmonized with future projections according to the Shared Socioeconomic Pathways (2010-2100). GFZ Data Services. https://doi.org/10.5880/pik.2017.007. This reference should be cited in the manuscript. RELATE TO THIS POINT LATER on doubts of data quality.(Geiger, 2018)

Thank you for introducing this data set. We have carefully gone through this page, and cite it in the introduction.

Q10) Section 2.3.4 (starting from line 335): It's worthwhile to acknowledge how policy changes can affect the projections and I agree that China's policy change regarding the two-children policy will have significant impacts. But I doubt that deliberately replacing the GDP estimates for one country only provide an improvement for a globally-consistent and dynamically-modelled dataset. On the contrary, China's economy is globally interconnected and simply adjusting China's output by as much as 40% will misrepresent the interaction with all other countries. There are two solutions: 1) stick with the original data provided or 2) wait for a new update of globally-consistent GDP projections that do not only include policy changes in China but similar changes across the globe including past economic crises and the Covid-19 pandemic.

We are more prone to the original plan of using population projections in China under two-children policy in GCP production. This policy has already been implemented and would no doubt affect the global population and GDP projection under all five SSPs. The original projections in the SSP database are already biased, and our GCP data set can at least provide an alternative choice of high resolution until new projection are made. Thank you.

Q11) Section 3.1.: I see many different data sources being used and reshuffled to obtain GDP per capita estimates. Relating to my major comment 8), I doubt that all GDP sources can be easily transferred between different units (in particular sub-national estimates used here). I further miss a thorough consistency check of GDP and

population data from different sources, including strange and un-referenced sources like the Wire & Plastic Products Group. Differences between different sources can be quite significant (e.g. because of different and/or changing assumptions on country shapes, …) and erroneous estimates of GDP per capita are the result. These issues are discussed for gridded population (Section 3.2.1) but totally neglected for national estimates here.

Thank you for this comment. We have given up the original inputs, and adopted GDP from Geiger et al. (2017) and GRP described in section 3.4 (and in Q7) as inputs.

The original data processing procedure is indeed quite confusing since we didn't give a detailed step by step instructor in the manuscript. This mistake has been carefully revised in our new manuscript. Please refer to sections 3.5 and 3.6.

Q12) Line 412-414: I find it quite strange to use a county-level analysis for China only to judge on a global population dataset quality. When using Fig S1 one would reject LandScan based on its largest RMSE contribution.

LandScan data set was chosen as final base map due to its high accuracy at a fine-scale. The figure S1(a) shows that three data sets in 2005 can well estimate population at a national scale. The relatively high RMSE of 5.1 million is due to one country with population totals of 1147.61 million from World Bank, but 1079.2 million from LandScan, 1094.7 million from WorldPop, and 1140.6 million from GPWv4.

[Figure]

Figure S1 Comparison between national (a) and subnational (b) population totals between official census against gridded data sets in 2005, values in the legend are their corresponding RMSE.

Comparison between 1564 states across the globe (Figure S1b) shows similar conclusion that these three data sets can well estimate population totals at a state scale, since population totals are openly available at state scale and have been used in population disaggregation for these three gridded data sets.

Since population totals at county scale are, however, not open accessed across the globe. Census in USA can be easily obtained, but not openly available in China. The comparisons in USA all show high accuracy for three population data sets, while in China, differences in accuracy can be obvious. So LandScan population data set was chosen here.

Thank you for this comment.

Q13) Eq 3: This approach is very general and does not refer to nor tests recent advancements in the field, see Eberenz et al (2020).

Thank you again for introducing the work from Eberenz et al (2020). LitPop approach has been carefully described and used in our revised manuscript.

Q14) Figure 2: I would be curious to see how the methodology applies to the U.S. or China alone instead of mixing up those countries. I would also strongly suggest to estimate the quality of the downscaling using log-transformed data. It seems in Fig 2c that the results are dominated by a few counties with largest GDP and not by the bulk of data.

Thank you for your suggestion. The evaluation has been carefully conducted and presented in section 3.4. National GDP in CURRENT PPP USD of 205 countries from World Bank was used only in GDP disaggregation here to evaluate their disaggregation

skills. Then we use official GDP and GRP in CURRENT PPP USD at national, state (476 in total from 38 countries), and county (5231 in USA and China) scales as validation, GCP using LitPop approach shows clear advantage in 2005 (Figure 1). And similar result can be obtained for 2015 (Figure S3).

Please see details in section 3.4.

[Figure]

Figure 1 Evaluation of official and reaggregated GDP (GRP) at national (a), state (b) and county (c) scales in 2005, using population-based disaggregation method, NTL-based approach, and LitPop approach, respectively. The GDP and GRP figures are in PPP current USD.

[Figure]

Figure S3 Evaluation of official and reaggregated GDP (GRP) at national (a), state (b) and county (c) scales in 2015, using population-based disaggregation method, NTL-based approach, and LitPop approach, respectively. The GDP and GRP figures are in PPP current USD.

It should be mentioned that all GDP (Geiger et al. (2017)) and GRP in 2005 PPP USD available (5932 administrative regions in total) is used in final GCP data

production, which shows high accuracy in more than 99% of administrative regions with bias less than 5%. Detailed information is presented in section 3.5.

[Figure]

Figure 2 Evaluation of official and reaggregated GDP and GRP in 2005 PPP USD of 5932 administrative regions across the globe in 2005. The reaggregated GDP comes from GCP using LitPop approach, and bias in (b) was calculated using $(GDP_{reaggregated} - GDP_{official})/GDP_{official}$

Q15) Section 3.4: The discussion is very general, not convincing and does not provide strong evidence to believe in the methodology presented. Again using only evidence from China and the U.S. for a global study provides very limited evidence, without discussing these limitations at all.

Thank you for this suggestion. The discussion has been largely improved. Advantages and limitations of LitPop approach have been carefully address in revised manuscript. Meanwhile, global and regional evidences have shown the superiority of this LitPop approach.

Q16) Line 508: See major comment 9.

Done. Thank you.

Q17) Line 510: This assumption is very strong for a period of about 80 years and could be overcome by either using the available downscaled SSP population data on 1km or by using more advanced refinements, as e.g. presented by Murakami and Yamagata 2019.

The population projections at 0.125 degree (Jones and O'Neill, 2016) have been used in GDP disaggregation. The LandScan population data set was used to further disaggregate into 1 km resolution for its spatial pattern. The downscaled population at 1 km was used in the first place, but then given up, as explained in Q8. Projections are available under SSP1-3 from Murakami and Yamagata (2019), with SSP4-5 missing, which limits the application here.

Thank you for this comment.

Minor comments:

Line 24: Definition ScenarioMIP missing

Done.

Line 28-31: This sentence is unclear and does not provide insights. What is not available open access?

Revised and remove from the manuscript. Thank you.

Line 66: Reference Tobias 2018 should be corrected to read Geiger 2018

Thank you and done.

Line 78: What are DN values? Please specify!

"DN" is abbreviation to "Digital Number", which is the brightness value of remote sensing images.

We have removed such description to increase the legibility of this manuscript. Thank you.

Line 95-98: I don't understand what the authors want to imply with this sentence.

Thank you for this comment. This sentence was meant to address the difficulty in making direct GCP predictions of using limited socioeconomic indicators as input, and GDP redistribution is the best choice to produce GCP projections for all five SSPs. This sentence has been revised to make it more clear.

*"Besides, most socioeconomic development indicators like the total factor productivity, capital stock, and labor input etc., are either short of data sources or provided mainly at a national scale, making it impossible to make direct GCP predictions, and GDP redistribution is the best choice to produce GCP projections for all five SSPs."*

Line 114: What gives rise to the range of projected GDP changes in China? Please specify!

Revised and done. Thank you.

*"Jiang et al., (2017; 2018) have updated Chinese population and GDP projections at provincial level that are qualitatively consistent with five SSP narratives, showing that the implementation of two-children policy can increase Chinese GDP by 38.1% – 43.9% in the late 21st century (Huang et al., 2019)"*

Line 377: Please provide reference to Wire & Plastic Products Group data sources.

Removed from the manuscript since this data was no longer used. Thank you.

Line 395: Please specify and reference which GIS-based boundaries were used in this study as this choice may have strong implications for the provided data.

Done and thank you. The shapefile was obtained from https://www.resdc.cn/data.aspx?DATAID=205.

Eq 3: The subscript lit-pop and NTL-Pop are used simultaneously, please correct.

Thank you, and done.

**The authors couldn't thank you enough for all the helpful and constructive suggestions and comments that improved our original manuscript. Very professional and we have gained more than we deserve. There is not a simple "thank you" that can express our gratitude no matter what happens next.**

**Hope all the best.**